# Unlabeled Abdominal Multi-organ Image Segmentation Based on Semi-supervised Adversarial Training Strategy

YuanKe Pan[#], Jinxin Zhu[#], and Bingding Huang[✉]

College of Big Data and Internet, Shenzhen Technology University, Shenzhen, 518188, China

**Abstract.** The unlabeled images are helpful to generalize segmentation models. To make full use of the unlabeled images, we develop a generator-discriminator training pipeline based on the EfficientSegNet, which has achieved the best performance and efficiency in previous FLARE 2021 challenge. For the generator, a coarse-to-fine strategy is used to produce segmentations of abdominal organs. Then the labeled image and the ground truth are applied to optimize the generator. The discriminator receives the original unlabeled image or the relevant noised image, together with their generated segmentation results to determine which segmentation is better for the unlabeled image. After the adversarial training, the generator is used to segment the unlabeled images. On the FLARE 2022 final testing set of 200 cases, our method achieved an average dice similarity coefficient (DSC) of 0.8497 and a normalized surface dice (NSD) of 0.8915. In the inference stage, the average inference time is 11.67 seconds per case, and the average GPU (MB) and CPU (%) consumption per case are 311 and 225.6, respectively. The source code is freely available at https://github.com/Yuanke-Pan/Adversarial-EfficientSegNet.

**Keywords:** Unlabeled image segmentation · Semi-supervised learning · Adverarial training

## 1 Introduction

The lack of labeled images is a great burden in medical imaging tasks. In the past five years, there are many researches [1,10,12] show the potential of semi-supervised algorithms in medical applications. In this paper, we focus on building an abdominal multi-organ segmentation model with only 50 labeled and 2000 unlabeled abdominal CT scans. There are two main difficulties in this task: 1) how to use those 2000 unlabeled CT scans to obtain more generalized results. 2) how to keep GPU memory and computation cost at a low level.

---

[1] #These authors contributed equally to this work.
[2] Corresponding author: Bingding Huang(huangbingding@sztu.edu.cn).

The EfficientSegNet [11] has shown excellent performance and efficiency in FLARE 2021 challenge [6]. The basic idea of our method is inheriting the network architecture of EfficientSegNet and applying semi-supervised learning algorithms to improve the generalizability of the inference model. In this work, we propose a novel adversarial strategy to use the unlabeled CT images which can improve the segmentation results. The main contributions of this work are:

1. We propose an adversarial semi-supervised learning pipeline for segmentation of medical images.
2. We design an image-label fusion discriminator to make full use of unlabeled images.

## 2   Method

### 2.1   Preprocessing

We use the same preprocessing strategy in both training and prediction. The details are described as follows:

- Reorienting the images to the left-posterior-inferior (LPI) view by flipping and reordering.
- Resampling the images to the fixed sizes. The sizes of coarse and fine input are [160, 160, 160] and [192, 192, 192], respectively.

  For labeled images, we implement the following preprocessing steps:

- Reorienting the masks to the LPI view by flipping and reordering.
- Cropping the images according to the masks with a margin of [20, 20, 20].

### 2.2   Proposed Method

In this work, we combined efficient context-aware network as generator and image-label fusion discriminator to make cross learning between labeled CT images and unlabeled images. The whole workflow of our method is illustrated in Fig. 1,

The training phase of our proposed adversarial semi-supervise learning workflow consists of three steps in every iterations:

- In step 1, we use the generator to produce segmentation result of the original image. Then we train the discriminator with the original image, the generated segmentation and the ground truth. Here we assume that the ground truth is always the best segmentation of the original image.
- In step 2, we train the generator with original image and the ground truth as supervised learning.

– In step 3, we first add the Gaussian noise to the unlabeled image to get the relevant noised image. Then we input the unlabeled image and the noised image into the generator to produce segmentation-Unlabeled and segmentation-Noised, respectively. After that, we input the unlabeled image and the noised image separately with the two generated segmentation results (segmentation-Unlabeled and segmentation-Noised) into the discriminator to determine which segmentation is better. The better segmentation will be used as the pseudo label and will be used to train the generator.

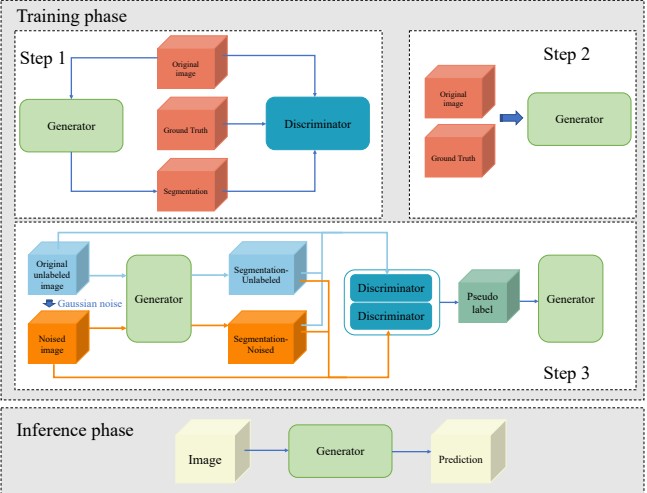

**Fig. 1.** The whole workflow of our proposed method. The first step is used to train the discriminator, the second step is used to train the generator, and the third step is used to process unlabeled data. Loop through three steps over multiple iterations.

It is important to emphasize that for the first 50 and the last 50 training iterations, we only use step 2 to fine-tune the network. After the training process, the final generator is used to segment the unlabeled images.

**Generator** For the coarse-to-fine training strategy, there are two different networks to extract global and local information. The coarse model is a typical 5-layer U-net structure which is employed to locate target organs. The fine model is the EfficientSegNet shown in Fig. 2, which receives the coarse segmentation results from the coarse model for further refinement.

**Discriminator** The input of the discriminator includes the original image and two segmentation results. For the labeled images, these two segmentation results are the generated segmentation and the ground truth. For the unlabeled images, these two segmentation results are derived from the unlabeled image and

the relevant noised unlabeled image. The discriminator is used to decide which segmentation is better for the original image.

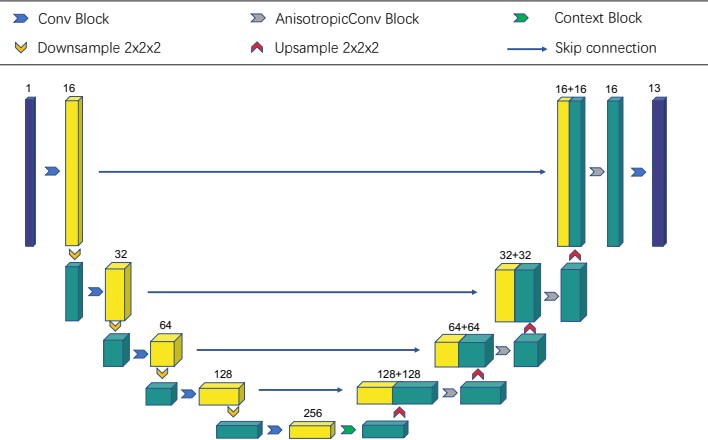

**Fig. 2.** The architecture of the fine network in the generator. The fine network performs fine segmentation on the coarse segmentation results of the corse network, and applies anisotropic convolution block to speed up the inference process.

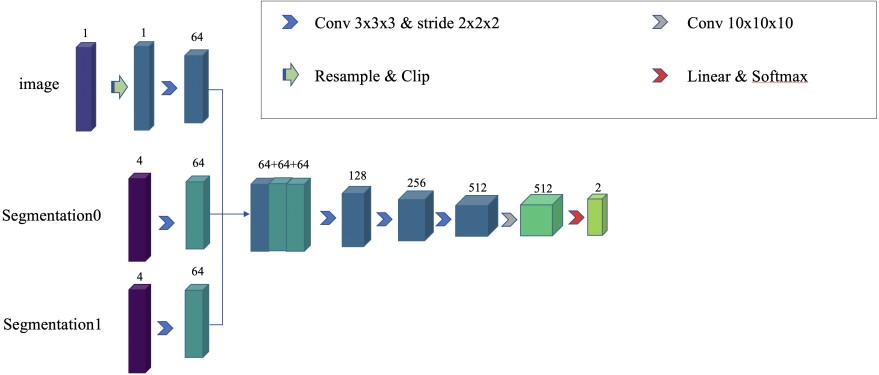

**Fig. 3.** The architecture of the discriminator network. Discriminator can be regarded as a condition-based binary classification model, which judges which segmentation is better according to the image.

As shown in Fig. 3, the image-label fusion discriminator fuses the original image with the two segmentation labels to output which label is better.

**Optional Operation** In order to improve the discriminator's performance, we apply an augmentation to the ground truth called mask poisoning. This augmentation step adds a random noise to the ground truth, which can make the discriminator more concentrate on the data distribution.

**Accelerate strategy** The same as the EfficientSegNet, the anisotropic convolution, anisotropic pooling and coarse-to-fine strategy are used to reduce inference time and GPU memory usage.

### 2.3  Post-processing

In the last step of the coarse-to-fine model, we apply a connected components analysis [8] to get the final segmentation results.

## 3  Experiments

### 3.1  Dataset and evaluation metrics

The FLARE 2022 dataset is curated from more than 20 medical groups under the license permission, including MSD [9], KiTS [3,4], AbdomenCT-1K [7], and TCIA [2]. The training set includes 50 labeled CT scans with pancreas disease and 2000 unlabeled CT scans with liver, kidney, spleen, or pancreas diseases. The validation dataset includes 50 CT scans with liver, kidney, spleen, or pancreas diseases. The test dataset includes 200 CT scans where 100 cases with liver, kidney, spleen, or pancreas diseases and the other 100 cases with uterine corpus endometrial, urothelial bladder, stomach, sarcomas, or ovarian diseases. All the CT scans only have image information and the center information is not available.

The evaluation metrics are two accuracy measurements: Dice Similarity Coefficient (DSC) and Normalized Surface Dice (NSD), and three running efficiency measurements: running time, area under GPU memory-time curve, and area under CPU utilization-time curve. All measurements will be used to compute the ranking. Moreover, the GPU memory consumption has a tolerance of 2 GB.

### 3.2  Implementation details

**Environment settings** The development environments and requirements are shown in Table 1.

**Training protocols** During the training process, we apply random shift and brightness to the whole dataset, and Gaussian noise to the unlabeled images. The details of the training protocols are shown in Table 2 and Table 3.

**Table 1.** Development environments and requirements.

| | |
|---|---|
| Windows/Ubuntu version | Ubuntu 20.04.1 LTS |
| CPU | AMD EPYC 7742 64-Core Processor |
| RAM | 16×4GB |
| GPU | coarse: 2*NVIDIA A100 40G 
 refine: 4*NVIDIA A100 40G |
| CUDA version | 11.2 |
| Programming language | Python 3.9 |
| Deep learning framework | Pytorch (Torch 1.11, torchvision 0.12.0) |

**Table 2.** Training protocols for the coarse model.

| | |
|---|---|
| Network initialization | "he" normal initialization |
| Batch size | Labelled data: 4 
 UnLabelled data: 2 |
| Patch size | 160×160×160 |
| Total epochs | 300 |
| Optimizer | Adam ($\mu = 0.99$) |
| Initial learning rate (lr) | 0.001 |
| Lr decay schedule | halved by 200 epochs |
| Training time | 6 hours |
| Loss function | $\mathbf{L} = 1 - \frac{2*\sum p_{\text{true}} * p_{\text{pred}}}{\sum p_{\text{true}}^2 + \sum p_{\text{pred}}^2}$ |

**Table 3.** Training protocols for the fine model.

| | |
|---|---|
| Network initialization | "he" normal initialization |
| Batch size | Labelled data: 8 
 UnLabelled data: 4 |
| Patch size | 192×192×192 |
| Total epochs | 500 |
| Optimizer | Adam ($\mu = 0.99$) |
| Initial learning rate (lr) | 0.001 |
| lr decay schedule | halved by 200 epochs |
| Training time | 3.4 hours |
| Loss function | $\mathbf{L} = 1 - \frac{2*\sum p_{\text{true}} * p_{\text{pred}}}{\sum p_{\text{true}}^2 + \sum p_{\text{pred}}^2}$ |

## 4   Results and discussion

### 4.1   Qualitative results on the validation dataset

We compare our proposed method with the baseline model on 20 cases of validation set. The qualitative results are shown in Table 4.

**Table 4.** Qualitative results of baseline and baseline+SS(semi-supervised) on 20 cases of validation set.

| Organ | Baseline | | Baseline+SS | |
|---|---|---|---|---|
| | DSC | NSD | DSC | NSD |
| Liver | 0.936 | 0.915 | 0.961 | 0.955 |
| RK | 0.818 | 0.796 | 0.838 | 0.815 |
| Spleen | 0.920 | 0.899 | 0.941 | 0.932 |
| Pancreas | 0.762 | 0.848 | 0.816 | 0.892 |
| Aorta | 0.945 | 0.977 | 0.958 | 0.986 |
| IVC | 0.835 | 0.825 | 0.891 | 0.892 |
| RAG | 0.709 | 0.820 | 0.660 | 0.745 |
| LAG | 0.709 | 0.798 | 0.821 | 0.900 |
| Gallbladder | 0.676 | 0.658 | 0.753 | 0.751 |
| Esophagus | 0.798 | 0.884 | 0.806 | 0.877 |
| Stomach | 0.731 | 0.752 | 0.864 | 0.883 |
| Duodenum | 0.675 | 0.829 | 0.718 | 0.811 |
| LK | 0.889 | 0.879 | 0.893 | 0.893 |
| Average | 0.800 | 0.837 | 0.840 | 0.872 |

In the case of comparison with the ground truth, Fig. 4 and Fig. 5 show two examples with good segmentation results and two examples with poor segmentation results, respectively. Table 5 and Table 6 shows the corresponding DSC and NSD scores. For all 20 cases in the validation dataset, the average of DSC and NSD of our model are 0.840 and 0.872, respectively. At the same time, we observe that the segmentation results of our proposed model on Right Adrenal Gland (RAG) is the worst in all organs, with the average DSC and average NSD of 0.660 and 0.745, respectively. One reason may be that there are too many organs around it.

Meanwhile, we compare our proposed method with the baseline model on 50 cases of validation set based on DSC. The qualitative results are shown in Table 7.

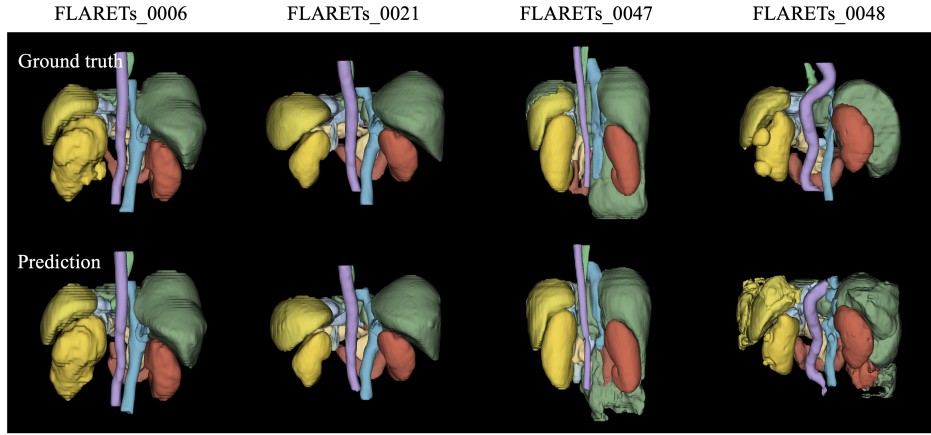

**Fig. 4.** Comparison of 3D segmentation in two good segmentation results (NO.6 and NO.21) and two poor segmentation results (NO.47 and NO.48).

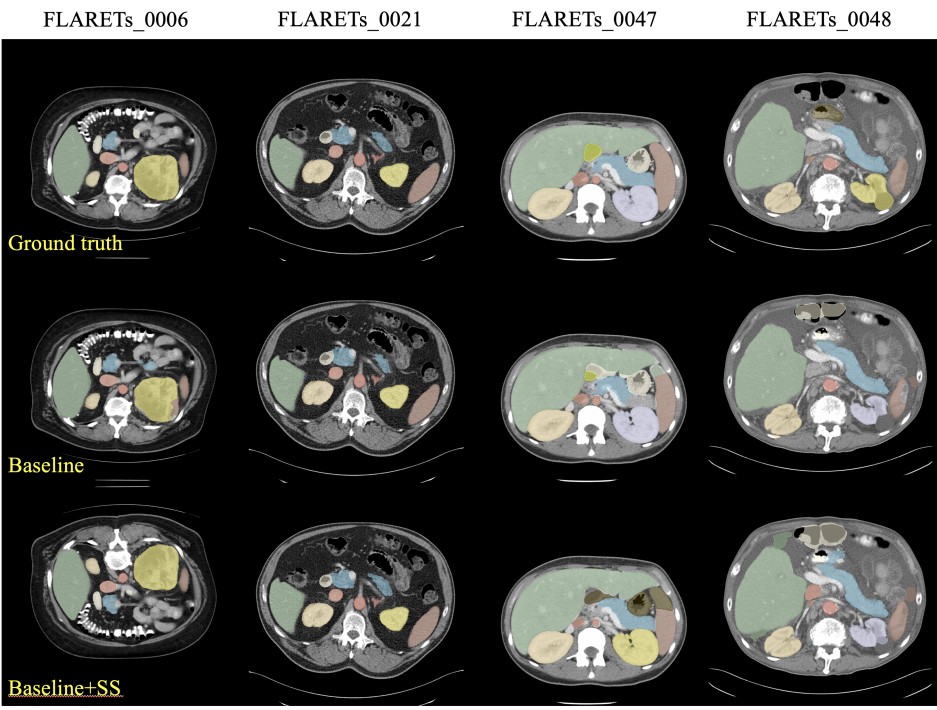

**Fig. 5.** Comparison of section segmentation of a layer in two good segmentation results (NO.6 and NO.21) and two poor segmentation results (NO.47 and NO.48).

**Table 5.** DSC of each organ in the two good segmentation results (NO.6 and NO.21) and two poor segmentation results (NO.47 and NO.48).

| Organ | NO.6 | NO.21 | NO.47 | NO.48 |
|---|---|---|---|---|
| Liver | 0.983 | 0.978 | 0.917 | 0.860 |
| RK | 0.979 | 0.978 | 0.984 | 0.738 |
| Spleen | 0.986 | 0.989 | 0.769 | 0.613 |
| Pancreas | 0.894 | 0.927 | 0.700 | 0.800 |
| Aorta | 0.976 | 0.972 | 0.938 | 0.828 |
| IVC | 0.970 | 0.951 | 0.734 | 0.523 |
| RAG | 0.898 | 0.807 | 0.000 | 0.788 |
| LAG | 0.926 | 0.888 | 0.860 | 0.921 |
| Gallbladder | 1.000 | 1.000 | 0.557 | 0.000 |
| Esophagus | 0.907 | 0.951 | 0.841 | 0.260 |
| Stomach | 0.938 | 0.977 | 0.734 | 0.297 |
| Duodenum | 0.826 | 0.939 | 0.000 | 0.708 |
| LK | 0.968 | 0.986 | 0.984 | 0.920 |
| Average | 0.942 | 0.949 | 0.694 | 0.635 |

**Table 6.** NSD of each organ in the two good segmentation results (NO.6 and NO.21) and two poor segmentation results (NO.47 and NO.48).

| Organ | NO.6 | NO.21 | NO.47 | NO.48 |
|---|---|---|---|---|
| Liver | 0.982 | 0.991 | 0.812 | 0.781 |
| RK | 0.988 | 0.986 | 0.998 | 0.658 |
| Spleen | 0.989 | 1.000 | 0.638 | 0.510 |
| Pancreas | 0.947 | 0.991 | 0.758 | 0.851 |
| Aorta | 0.998 | 0.999 | 0.988 | 0.850 |
| IVC | 0.996 | 0.962 | 0.688 | 0.581 |
| RAG | 0.968 | 0.918 | 0.000 | 0.888 |
| LAG | 0.989 | 0.972 | 0.944 | 0.984 |
| Gallbladder | 1.000 | 1.000 | 0.610 | 0.000 |
| Esophagus | 0.988 | 1.000 | 0.892 | 0.335 |
| Stomach | 0.948 | 0.999 | 0.772 | 0.373 |
| Duodenum | 0.871 | 0.999 | 0.001 | 0.836 |
| LK | 0.934 | 0.998 | 0.985 | 0.876 |
| Average | 0.969 | 0.986 | 0.699 | 0.656 |

**Table 7.** Qualitative DSC results of baseline and baseline+SS (semi-supervised) on 50 cases in the validation dataset.

| Organ | baseline+SS | baseline |
|---|---|---|
| Liver | 0.9555 | 0.9150 |
| RK | 0.9021 | 0.8586 |
| Spleen | 0.9341 | 0.8603 |
| Pancreas | 0.8069 | 0.7422 |
| Aorta | 0.9487 | 0.9180 |
| IVC | 0.8762 | 0.8065 |
| RAG | 0.6678 | 0.6842 |
| LAG | 0.7297 | 0.6511 |
| Gallbladder | 0.7644 | 0.6652 |
| Esophagus | 0.8018 | 0.7412 |
| Stomach | 0.8800 | 0.7279 |
| Duodenum | 0.7298 | 0.6588 |
| LK | 0.9066 | 0.8663 |
| Average | 0.8387 | 0.7766 |

### 4.2    Segmentation efficiency results on validation set

Our docker was validated with NVIDIA QUADRO RTX5000 (16G) and 32G RAM on 50-case validation set. The mean running time per case is 11.6 seconds, the mean maximum GPU memory is 2423MB, the mean area under GPU memory-time curve is 12287 and the mean area under CPU utilization-time curve is 224.

### 4.3    Results on final testing set

On the final testing set of 200 cases with undisclosed ground truth, the average DSC and NSD of our method were 84.97 and 89.15, respectively. In terms of segmentation efficiency, the mean running time per case is 11.67 seconds, the mean area under GPU memory-time curve is 12155 and the mean area under CPU utilization-time curve is 226. The average DSC and NSD for these 13 organs are shown in Table 8.

### 4.4    Limitation and future work

In this work, we separate the training pipeline into three steps. In each step, the parameters of the discriminator and the generator are not updated at the same time, it might cost additional training time. An end-to-end semi-supervised learning algorithm will be our goal in the future. Due to the time limitations, we do not find a way to perform the proper analysis of the dataset. Considering the success of nn-UNet [5], we believe that a proper analysis might improve our final segmentation results.

**Table 8.** Average DSC and NSD of 13 organs on the final test set.

| Organ | DSC | NSD |
|---|---|---|
| Liver | 0.9650 | 0.9653 |
| RK | 0.9166 | 0.9075 |
| Spleen | 0.9307 | 0.9329 |
| Pancreas | 0.7721 | 0.8708 |
| Aorta | 0.9583 | 0.9796 |
| IVC | 0.8688 | 0.8727 |
| RAG | 0.7698 | 0.8804 |
| LAG | 0.7838 | 0.8794 |
| Gallbladder | 0.7705 | 0.7740 |
| Esophagus | 0.7590 | 0.8405 |
| Stomach | 0.8840 | 0.9040 |
| Duodenum | 0.7280 | 0.8474 |
| LK | 0.9396 | 0.9346 |
| Average | 0.8497 | 0.8915 |

## 5    Conclusion

In this paper, in order to use the large amount of unlabeled data, we develop an adversarial generator-discriminator training pipeline based on EfficientSegNet. For the generator, we employ the coarse-to-fine strategy to generate segmentation results. The labeled images and their ground truth are used to optimize the generator. For the discriminator, we first add noise to the original unlabeled images, then the discriminator receives the original unlabeled image or the relevant noised image together with their generated segmentation results. The better segmentation is determined by the discriminator as the pseudo label of the unlabeled image. On the FLARE 2022 validation dataset, our method achieved an average DSC of 0.840 and a NSD of 0.872 with an average process time of 11.6 seconds per case in the inference phase.

**Acknowledgements** The authors of this paper declare that the segmentation method implemented for participation in the FLARE 2022 challenge has not used any pre-trained models nor additional datasets other than those provided by the organizers. The proposed solution is fully automatic without any manual intervention.

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
