# OpenReview forum: "Unlabeled Abdominal Multi-organ Image Segmentation Based on Semi-supervised Adversarial Training Strategy"
_MICCAI.org/2022/Challenge/FLARE_

### Official Review · Reviewer_aLiY · 2022-09-21
**Unlabeled Abdominal Multi-organ Image Segmentation Based on Semi-supervised Adversarial Training Strategy**

**Rating:** 8
**Confidence:** 4

**Review:**

Pros:
- an adversarial generator-discriminator training pipeline based on EfficientSegNet

Cons:
- lack average used GPU memory

---

> ### Author Response · Authors · 2022-10-10
> **Many thanks for the reviewer's attention and suggestions. This is helpful for perfecting our paper.**
>
> Many thanks for the reviewer's attention and suggestions. This is helpful for perfecting our paper.
> - lack average used GPU memory
>
>     we add Section 4.2 and modify Section 4.3 to show the results.
> Looking forward to your more attention and suggestions.

---

### Meta-Review · Program_Chairs · 2022-09-30

**Recommendation:** Major Revision
**Confidence:** 5

**Metareview:**

Reviewer 2:
1. Please include the ORCID for all authors.
2. Please provide the results w/ and w/o unlabeled data on the validation set (50 cases).
3. Please analyze the segmentation efficiency
4. Please adjust the window size and level of CT images to 400/40 in Fig. 5
5. Fig. 4-5 can be zoomed in to fill the space.

Reviewer 3:
1. The authors should attach ORCID if possible.
2. The segmentation efficiency analysis is missed.
3. Please add the license to your code repo.
4. Please add detailed explanations of the networks in the titles of Figures 1-3

Meta review：

Reviewers raise many concerns and suggestions. Please address all comments in the revised manuscript.

---

> ### Author Response · Authors · 2022-10-10
> **Thanks to the reviewers for their many concerns and suggestions, we give the corresponding solutions**
>
> Many thanks for the reviewer's attention and suggestions. This is helpful for perfecting our paper.
>
> #### Reviewer 2:
>
> 1. Please include the ORCID for all authors.
>
>    We added the author's ORCID to the article.
>
> 2. Please provide the results w/ and w/o unlabeled data on the validation set (50 cases).
>
>    we add Table 7 to show the corresponding results.
>
> 3. Please analyze the segmentation efficiency
>
>    we add Section 4.2 and modify Section 4.3 to show the segmentation efficiency.
>
> 4. Please adjust the window size and level of CT images to 400/40 in Fig. 5
>
>    We have complied.
>
> 5. Fig. 4-5 can be zoomed in to fill the space.
>
>    We have complied.
>
> #### Reviewer 3:
>
> 1. The authors should attach ORCID if possible.
>
>    We added the author's ORCID to the article.
>
> 2. The segmentation efficiency analysis is missed.
>
>    we add Section 4.2 and modify Section 4.3 to show the segmentation efficiency.
>
> 3. Please add the license to your code repo.
>
>    We have complied.
>
> 4. Please add detailed explanations of the networks in the titles of Figures 1-3
>
>    We have complied.
>
> Looking forward to your more attention and suggestions.